# Perspectives on the Application of Cytogenomic Approaches in Chronic Lymphocytic Leukaemia

**DOI:** 10.3390/diagnostics13050964

**Published:** 2023-03-03

**Authors:** Wan Norizzati Wan Mohamad Zamri, Nazihah Mohd Yunus, Ahmad Aizat Abdul Aziz, Ninie Nadia Zulkipli, Sarina Sulong

**Affiliations:** 1Human Genome Centre, School of Medical Sciences, Universiti Sains Malaysia, Kubang Kerian 16150, Malaysia; 2School of Biomedicine, Faculty of Health Sciences, Universiti Sultan Zainal Abidin, Kuala Terengganu 21300, Malaysia

**Keywords:** cytogenomics, chronic lymphocytic leukaemia, microarray

## Abstract

Chronic lymphocytic leukaemia (CLL) is a haematological malignancy characterised by the accumulation of monoclonal mature B lymphocytes (positive for CD5+ and CD23+) in peripheral blood, bone marrow, and lymph nodes. Although CLL is reported to be rare in Asian countries compared to Western countries, the disease course is more aggressive in Asian countries than in their Western counterparts. It has been postulated that this is due to genetic variants between populations. Various cytogenomic methods, either of the traditional type (conventional cytogenetics or fluorescence in situ hybridisation (FISH)) or using more advanced technology such as DNA microarrays, next generation sequencing (NGS), or genome wide association studies (GWAS), were used to detect chromosomal aberrations in CLL. Up until now, conventional cytogenetic analysis remained the gold standard in diagnosing chromosomal abnormality in haematological malignancy including CLL, even though it is tedious and time-consuming. In concordance with technological advancement, DNA microarrays are gaining popularity among clinicians as they are faster and better able to accurately diagnose the presence of chromosomal abnormalities. However, every technology has challenges to overcome. In this review, CLL and its genetic abnormalities will be discussed, as well as the application of microarray technology as a diagnostic platform.

## 1. Introduction

Chronic lymphocytic leukaemia (CLL) is a chronic lymphoproliferative disorder characterised by accumulation of mature monoclonal B lymphocytes, more than 5000 per microlitre in peripheral blood, positive for immunophenotype marker (CD5+ and CD23+) and/or the involvement of lymph nodes [1]. It is a common type of leukaemia in adults, especially in Western countries. The estimated incidence of this disease in the Western population (USA and Europe) is approximately 5 new cases per 100,000 individuals, regardless of gender [2]. In the USA itself, the estimated number of newly diagnosed cases for 2020 was 21,040 cases, which was around 1.2% of all cancer cases. The median age at diagnosis of this disease is 72 years old [3,4,5]; there is male predominance with a male-to-female ratio of approximately 2:1 [6,7,8]. It accounts for about 1% to 3% from total non-Hodgkin lymphoma cases reported. In contrast, the CLL cases reported in Asian countries as well as East Asia (0.1–0.2/100,000) [9,10], Africa (0.66/100,000) [11], and South America (Hispanic descendants) (1.17/100,000) [12] are relatively low compared to their Western counterparts [13,14,15,16,17,18,19,20]. Further, in Japan, CLL is classified as a rare disease, with the reported incidence rate being far below 0.5 per 100,000 person-years [21,22,23,24]. It is challenging to diagnose CLL in Japan due to the disease’s high degree of morphological and immunological variability [25]. For Australia and New Zealand, CLL is considered a common type of leukaemia to be diagnosed with, having an incidence rate of 2.99 per 100,000 [26]. In contrast, CLL is a rare disease in Africa [27,28]. Just 40 patients, with an average age of 61, were diagnosed with CLL over the course of 3 years across many centres in Senegal [29]. Meanwhile, in 2019, the reported incidence rate of CLL in Central Latin America was 0.41 per 100,000 individuals. Since the incidence rate of CLL in 1990 was 0.28 per 100,000, the incidence rate reported in 2019 has grown [26]. According to the Malaysia National Cancer Registry Report 2007–2011, the total number of new cases of CLL in Malaysia from 2007 to 2011 was 124 patients, implying that there were 24.8 newly diagnosed CLL cases per year on average [30]. This disparity in cases reported suggests that Asian CLL has different biological characteristics and, in some cases, has different chromosomal abnormalities when compared to Western CLL [31,32,33,34,35]. This disparity in disease incidence is postulated as being related to genetic differences between races.

Although most of the CLL cases are asymptomatic and usually managed with watching-and-waiting until development of symptoms occurs—such as cytopenia, lymphadenopathy, and splenomegaly—in some patients, it will transform into aggressive form of B lymphocyte malignancy such as diffuse large B cell lymphoma (DLBCL) or, rarely, transform into Hodgkin lymphoma or another type of aggressive lymphoma [1].

## 2. Genetics of CLL

CLL is a heterogeneous disease. Its pathogenesis can be viewed as cooperation between a patient’s risk factors and genetic aberrations. There have been several studies performed to identify risk factors for CLL development; however, to date, there is still no specific acquired factor that has been identified for disease development. However, there is strong evidence that genetic predisposition can lead to CLL [36,37,38,39]. Host factors including family history with haematological malignancy (CLL and/or non-Hodgkin lymphoma (NHL)) are among the strong evidence that has been studied. The study performed by Slager et al. revealed that relatives of CLL patients have a 2- to 8-fold increase in the risk of developing CLL and a 2-fold increased risk of getting NHL compared to the general population [40]. This finding was also supported by those of Goldin et. al, which state that familial CLL was diagnosed at an earlier age compared to sporadic CLL [41]. There are also case reports involving familial CLL where two or more individuals were affected by CLL in the same family.

Histone modifications, such as those linked to active enhancer and promoter elements and regions of the genome that were actively transcribed, have been shown to play a role in the epigenetics of CLL. Additionally, it has been discovered that single-nucleotide polymorphisms (SNP), which increase the risk of CLL, overexpress transcription factor binding [42]. The latest studies using genome-wide association studies (GWAS) revealed more than 40 susceptibility loci which were important in B lymphocytes and apoptotic pathways [42,43].

The common chromosomal aberrations associated with CLL are del 13q14, trisomy 12, del 11p, and del 17p [44,45,46,47]. Other chromosomal aberrations observed in CLL are deletions in 6q, 9p21, and 10q23, total or partial trisomy of chromosomes 3, 8, 18, or 19, and duplications in 2p24 [47,48,49,50]. The most common genetic lesions in CLL are deletions of 13q14 (del 13q14), generally monoallelic in 50~60% patients (Figure 1; Table 1) [18], and involve the deletion of regions containing two long non-coding RNA genes (*DLEU2* and *DLEU1*) which later develop clonal lymphoproliferation, recapitulating the different steps of CLL initiation and progression. Deletion 13q14 causes dysregulation of microRNAs, i.e., miR15A and miRNA16A, which are encoded in the deleted region. Both microRNAs have critical roles in controlling the production of proteins essential for cell apoptosis and normal cell cycle progression [51]. Consequently, cells are unable to respond to stress signals in a way that promotes apoptosis and leads to disease progression when these microRNA regions are absent [52]. Besides that, deletion of miR16A and miR15A causes upregulation of the BCL2 gene in CD5+ cells, which activates the BCL-2 proto-oncogene aberrant signalling pathway and assist in the development of the disease [53]. Deletion 13q14 is associated with good prognosis as well as prolonged time to first treatment (TTFT) and prolonged overall survival compared to other genetic abnormalities [44,51].

Trisomy 12 is a chromosomal aberration in CLL found in 10–20% of cases and often appears as a unique cytogenetic alteration (40–60% of cases with trisomy 12) (Figure 1; Table 1). In addition, it can be associated with other chromosomal aberrations, including trisomy 18 and 19, recurrent CLL deletions (e.g., 14q, 13q, 11q, or 17p), and IGH translocations [54]. Trisomy 12 is also associated with an atypical morphology of the lymphocytes. Although trisomy 12 is considered an intermediate-risk genetic lesion in CLL, the co-occurrence with NOTCH1 mutations are associated with poor survival outcome [55]. This finding is also in line with the increased frequency of trisomy 12 in Richter syndrome patients.

Deletion of the long arm of chromosome 11 is detected in 5–20% of CLL patients (Figure 1; Table 1) [45,56,57]. This deleted region of chromosome 11 usually harbours ATM gene in almost all cases, as well as other genes including RDX, FRDX1, RAB39, CUL5, ACAT, NPAT, KDELC2, EXPH2, MRE11, H2AX, and BIRC3. ATM gene mutations have been largely studied in CLL patients with del(11q); however, they have been found in only 8–30% of 11q- patients [58,59], indicating that other genes could play a role in the pathogenesis of 11q deletions in CLL. One of these genes is BIRC3, which is located near to the ATM gene, at 11q22. BIRC3-disrupting mutations and deletions have been rarely detected in CLL at diagnosis (4%) but are detected in 24% of fludarabine-refractory CLL patients, suggesting that BIRC3 genetic lesions are specifically associated with a chemo-refractory CLL phenotype [60,61]. CLL patients with del(11q) are characterised by large and multiple lymphadenopathies and have been associated with progressive disease and poor prognostic factors, such as unmutated IGHV genes. It has been associated with shorter TTFT, shorter remission durations, and shorter OS following standard chemotherapy compared to non-deleted 11q (and non-deleted 17p) cases [62].

Deletion of 17p, especially at the region 17p13 chromosomal region (del17p), can be found at different frequencies depending on clinical stages of CLL disease, ranging from 1–3% during initial diagnosis to 20% in chemo-refractory disease (Figure 1; Table 1) [48,54]. Deletion 17p is associated with TP53 inactivation, thus causing genomic instability. This deletion is also linked to resistance to DNA-damaging agents (radiotherapy or chemotherapy) and presence at diagnosis usually indicates unfavourable OS and decreased TTFT.

**Table 1 diagnostics-13-00964-t001:** The common chromosome aberrations in CLL tested by FISH.

Chromosome Aberrations	Prevalence at Diagnosis	Gene Involved	Implication	Prognostic Risk	Ref.
Deletion 13q14	50–60%	DLEU2 and DLEU1	Clonal lymphoproliferation, recapitulating the different steps of CLL initiation and progression.	Good	[18]
Biallelic losses in 13q	Almost 30% of 13q-deleted CLL patients			It is speculated that the prognostic effect of biallelic mutations may be obscured by the magnitude of deletions or the silencing of the remainder allele through other processes.	[63]
Trisomy 12	10–20%		Associated with atypical morphology of the lymphocytes.	Intermediate	[49,55,63]
Deletion 11q22-23	5–20%	ATM	Associated with chemo-refractory CLL.	Poor	[45,56,57]
Deletion 17p13	1–3% (initial diagnosis) >20% (in chemo-refractory disease)	TP53	TP53 inactivation causing genomic instability and linked to resistance to radiotherapy and/or chemotherapy.	Poor	[48,54]

In addition to the common chromosome aberrations detected by FISH, Table 2 displays various chromosome abnormalities in CLL patients revealed by other platforms.

Robbe et al. (2022) identified 74 regions of the genome that were currently affected by copy number alterations (CNAs), including 14 well-known CNAs such as del13q14.2, del11q22.3, and del17p13.1, through microarray. Another 60 regions—of which, 27 were previously not recognised and the remaining 33 CNAs—could be refined to a smaller minimal overlapping region. The author also demonstrated the most likely target gene for nine known regions, includingTP53/del17p13.1, and seven additional regions, including *PCM1*/del8p, *IRF2BP2*/del1q42.2q42.3, and *SMCHD1*/del18p11.32-p11.31 [72].

Certain gene mutations, in addition to chromosomal abnormalities, are critical to CLL pathogenesis, and multiple subpopulations of evolving malignant cells have been identified. These modifications have an impact on intracellular or microenvironment-dependent signalling pathways [58]. Over 5% of CLL patients have mutations in NOTCH1, ATM, SF3B1, and TP53. Notch proteins regulate the development of haematopoietic cells by acting as cell transmembrane receptors. Mutations in NOTCH1 at proto-oncogenes’ coding and non-coding regions can worsen disease through splicing events and increase their overall activity [73]. ATM, as previously stated, is a gene that detects damaged DNA and induces cell apoptosis, and its mutation will lead to dysregulation of the cell cycle [74]. SF3B1 is the gene that produces nuclear ribonucleoproteins, which are required for messenger RNA splicing and, thus, affect the cell cycle [75]. As previously stated, TP53 is essential for responding to DNA damage and inducing cell apoptosis.

Aberrant signalling pathways also play important roles in the pathophysiology of CLL. The three main pathways involved are antigen-independent BCR signalling, BCL2 proto-oncogene upregulation, and impaired DNA damage response. Through antigen-independent or antigen-dependent autonomous signalling of CLL cells, the antigen-independent BCR signalling pathway directly affects cell survival, growth, differentiation, and cellular adhesion or migration. It is influenced by low miR150 levels as well as high FOXP1 and GAB1 expression [76]. BCR activation causes the kinases such as PI3K, SYN, BTK, and LYN to be activated, which results in cytoplasmic domain integrin activation and conformational changes that allow more ligand to bind to integrin’s extracellular activity, affecting cell proliferation, migration, differentiation, and survival [77]. Somatic mutations in immunoglobulin heavy chain variable region (IGHV) genes also affect the antigen-independent BCR signalling pathway. Mutated IGHV has weaker BCR signalling due to narrower antigen specificity, resulting in a higher mutation burden and a lower frequency of driver mutations. As a result, mutated IGHV CLL cells proliferate more slowly, making the disease process more benign and less clinically aggressive. Unmutated IGHV CLL cells, on the other hand, have sustained BCR signalling by binding to multiple epitopes, resulting in a lower mutation burden and a higher driver mutation frequency. This process eventually leads to faster clonal expansion and more clinically aggressive disease [75,78]. Table 3 highlights the gene mutations that contribute to the prognosis of CLL.

Even though the incidence of CLL in Western countries is higher than in Asian countries, the disease progression in Asian patients has been reported to be more aggressive and with a shorter time to treatment compared to its counterpart. This event was postulated to happen due to different biomarkers and susceptibility in Asian populations. Based on a prospective study conducted in Senegal by Sall et al., CLL was found to be more aggressive and had a poorer prognosis at a younger age than in developed nations [29]. To depict the exact pattern of disease progression in African countries, however, it was necessary to conduct large-scale epidemiological research in African countries, as this study only represents a small-scale African study [13]. There were several case reports showing Asian CLL had reported a few different chromosomal aberrations than Western CLL. Western and Asian CLL shared the major copy number changes, which are del13q14, trisomy 12, deletion 17p, and deletion 11q [79]. Kawamata et al. also reported that Asian CLL patients more frequently have either trisomy/duplication of 3q or trisomy 18/dup18q; none of these chromosomal aberrations were reported in Western CLL patients [80]. Another study performed by Wu and his team members revealed Asian CLL patient had high frequency of TP53 mutation compared to Western CLL [81,82]. Prior to the last two decades, it was reported that the common chromosome abnormalities of CLL in South Africa are comparable to those of the rest of the world [83]. In 2016, Sall and colleagues found that CLL patients in Senegal exhibited the same clinical presentation as individuals globally. The epidemiology of haematologic malignancies, particularly CLL, is less understood in Latin America (Central and South America) [84]. A study performed by Hahn and his colleagues discovered two gene candidates, *PRPF8* and *SAMHD1*, in Australian familial CLL [85]. Even though African CLL is considered rare, their patients usually have a younger median age of onset (59 years old), higher frequency of adverse prognostic factors, and poor clinical outcome. It also found that *TP53*, *SF3B1*, and *NFKBIE* mutations in African CLL is higher than in Western CLL [86].

For almost 40 years, the Rai and Binet clinical staging systems, which base their evaluations on a patient’s physical examination as well as their blood counts, have served as the foundation for determining a patient’s prognosis in CLL [87,88]. Rai and modified Rai classification stress the lymphocytes count and nodal and organ (spleen) involvement more, while Binet classification looks more at haemoglobin level, platelet count, and number of nodal areas involved. However, the information gained from these classifications during diagnosis of CLL in patients will not be able to predict the progression of disease in each individual [1]. Recently, an international team of researchers reviewed data from patients participating in eight randomised clinical trials in Europe and the United States in order to construct a prognostic score that contains widely available clinical, biochemical, and genetic prognostic characteristics. The CLL International Prognostic Index (CLL-IPI) was developed as a result of this international effort, and it is a reasonably straightforward prognostic tool. This prognostic model divides patients into four distinct categories, each of which has a significantly different overall survival rate, based on five parameters such as age, clinical stage, *TP53* status (normal vs. del(17p), and/or TP53 mutation), *IGHV* mutational status, and serum β2-microglobulin. Subsequently, the prognostic utility of the CLL-IPI was validated in two separate cohorts of newly diagnosed patients, one from the Mayo Clinic and the other from the Swedish CLL registry [89]. Despite the fact that CLL-IPI was initially established to predict overall survival, it was found that the index could also predict TTFT in newly diagnosed patients with CLL. Only 20% of the original dataset consisted of patients with early illness, and no effort has been made to optimise the CLL-IPI risk score to stratify TTFT among early-stage patients. It is important to emphasise that TTFT is a disease-specific goal that is more relevant than overall survival for patients who have recently been identified with early-stage disease [90,91,92]. The CLL-IPI is used as a supplement for the existing methods of risk stratification for CLL [93].

## 3. Cytogenomic Approaches in CLL: Advantages and Challenges

For decades, diagnosis of CLL was performed using a full blood picture with the presence of lymphocytes more than 5 × 10^9^/µL, examination of marrow morphology, marrow immunophenotyping, marrow cytogenetics, and clinical examination to detect nodal involvement. However, for the past 10 years, rapidly developed technology has made the detection of genetic aberrations in haematological malignancies, especially in CLL, become more comprehensive and elaborate. Genetic aberration detection plays a pivotal role in diagnosis, disease prognosis determination, risk stratification, and survival outcome. It is also essential in specific targeted therapy selection that is tailored to a patient’s genetic aberrations in order to achieve a better outcome [37]. Various methods of cytogenomic testing can help clinicians to detect the presence of genetic aberrations in patients.

Cytogenomics can be defined as the study of the numerical and structural variation of the genome at the chromosomal and subchromosomal level as well as at a molecular resolution using methods that cover the entire genome or specific DNA sequences [94,95]. It evaluates chromosomes and their relation to disease [96]. The term “cytogenomics,” also called “chromosomics,” was proposed by Uwe Claussen to highlight the three-dimensional morphological changes that occur in chromosomes and which are crucial aspects in the regulation of genes [97]. Cytogenomic testing is not limited to conventional cytogenetic analysis (CCA) and molecular cytogenomics methods, i.e., fluorescence in situ hybridisation (FISH), polymerase chain reaction (PCR), or Multiplex Ligation-dependent Probe Amplification (MLPA); it also comprises high-throughput cytogenomics technologies which include applications of whole-genome Copy Number Variation (CNV) analysis such as DNA microarray, next-generation sequencing (NGS), and, more recently, GWAS as a diagnostic method [98,99]. These fancy, sophisticated, and typically very costly methods are only possible in conjunction with high-tech apparatuses and/or bioinformatics. In return, they are competent for achieving a high-resolution view of genomes as well as the generation of massive data sets in a time-effective manner [100]. Furthermore, cytogenomics exemplify the understanding of genomic instability and its association with normal and abnormal aging throughout ontogeny which later may contribute to cancer development [101].

Until now, CCA still remains the gold standard to diagnose chromosomal aberrations in CLL, especially in detecting the presence of complex karyotypes or balanced chromosomal translocations [54,102,103]. However, CCA is time-consuming, unable to assess non-dividing cancer cells, and sometimes yields poor morphology or inadequate cells for analysis [104,105]. It also can only detect chromosomal aberrations around 30% of CLL cases [106,107]. In developed countries, this method has become the last choice as array-based testing is more favoured and CCA only acts as last resort in detecting balanced chromosomal abnormalities.

Based on a number of prospective clinical trials, the latest International Workshop on Chronic Lymphocytic Leukemia (iwCLL) guidelines for the management of CLL recommend performing FISH analysis as well as analysis of the TP53 gene in all patients with CLL, in both general practice and clinical trials. The use of CCA is recommended only in the context of clinical trials rather than routine clinical settings. This recommendation is mostly based on recent reports highlighting the prognostic significance of complex karyotype (CK) which, presently, can be detected only through CCA [94,108].

In CLL, CK is classically defined as the presence of ≥3 clonal structural or numerical abnormalities. Although present in 8% of monoclonal B lymphocytosis cases, 26 CK ≥3 is associated with advanced-stage disease, cases harbouring unmutated IGHV genes (U-CLL), del(11q), TP53 aberrations [del(17p) and/or TP53 mutation], and telomere dysfunction [109,110].

Combining FISH with NGS, as well as FISH and long-range sequencing methods, has led to significant advances in the field of cytogenomics in the 2010s [111,112]. FISH techniques are the most effective for researching genomes’ repetitive sections [113], and as a result, numerous probes targeting heterochromatic and euchromatic areas of the human genome have been created [111]. In early 2010, FISH and MLPA were becoming more popular as tools to diagnose chromosomal aberrations in CLL. However, despite the high sensitivity test for both methods, they are limited to specific known genomic loci [114,115]. FISH and MLPA act as the supplementary test to CCA. Both can be used in diagnosing genetic aberrations in non-dividing cells with high specificity and sensitivity. FISH is also able to detect low levels of mosaicism and mosaics of mono- and biallelic deletions [116,117]. However, FISH testing needs to be performed separately with specific probes for each genomic abnormality, making this method relatively expensive and time-consuming. It also unable to detect any other chromosomal abnormalities aside from the known genomic loci that have been specified by probes [47]. FISH is more sensitive than karyotyping; nevertheless, it is only effective for analysing specified loci, and it requires an assay for each targeted aberration [118]. While MLPA testing is able to detect copy number alterations, methylation pattern changes, and/or even point mutations simultaneously in multiple target regions [114,119,120,121], it has its own disadvantages. This method cannot detect copy-neutral loss of heterozygosity and has problems with mosaicism, i.e., unable to be obtained, tumour heterogeneity, or sometimes can cross contaminate with normal cells [116]. This finding proves that FISH and MLPA cannot be a stand-alone test and only able to act as complementary test for CCA.

The emergence of microarray-based comparative genome hybridisation (array-CGH) and high-density single-nucleotide polymorphism (SNP) arrays has led to deeper understanding of the CLL genomic landscape. By delivering a genome-wide, high-resolution analysis that does not require cell culturing or viable cells for testing, chromosomal microarray analysis fills the void between genome-wide low-resolution chromosome studies and region-limiting disease-specific targeted FISH panels [122,123]. However, array-CGH has a several shortcomings, including its inability to detect low-level mosaics, its insensitivity to heterochromatin, and its inability to detect balanced aberrations. Only copy number variations were able to be identified between the years 2000 and the 2010s [124,125,126]. Initially, microarray-based detection of copy number alterations (CNAs) is the standard of care for the diagnosis of most constitutional chromosomal imbalances in children with developmental disability abnormalities [123], but recently it has become more popular for diagnosing haematological malignancies. Microarray technology, especially that using CNA+SNP chip technology, is the best at diagnosing aneuploidies, microdeletions, especially cryptic loci deletion and duplications, as well as amplification in CLL. It also can detect additional confirmation of CNAs and the ability to detect copy-neutral loss of heterozygosity (CN-LOH) and some polyploidies. The integration of microarray analysis into the cytogenetic diagnosis of haematological malignancies improves patient management by providing clinicians with additional information about potentially clinically actionable genomic alterations [123]. However, every technology has its own limitation. Microarray limitation include the inability to detect balanced rearrangements, decreased performance at low levels of tumour [50], the need for well-trained laboratory technologists, and high operation costs, even though this method is far superior compared to CCA, FISH, and MLPA [127].

Examples of commonly used microarray platforms in haematological malignancies are the CytoScan HD array platform (Affymetrix) and the HumanOmniExpress Array (Illumina). Both platforms use CNA+SNP chip technology in detecting cytogenomic alterations. Data obtained by the CytoScan HD array platform supplied by Affymetrix were analysed using the Chromosome Analysis Suite software while HumanOmniExpress platform data were analysed using Nexus copy number software (Biodiscovery Inc.) using annotations of genome version GRCh37 (hg19). In a study done in the Netherlands by Steven-Kroef et. al, both platforms show a high limit of resolution and detection of clinically relevant genomic aberrations which were unable to be detected by CCA and FISH [127].

For the past few years, optical genome mapping (OGM) has emerged as a promising new approach that may be able to circumvent all of the aforementioned testing hurdles with a single, comprehensive analysis. OGM is based on high-throughput imaging of long DNA molecules (>250 Kb) that have been fluorescently labelled at a specific 6 bp sequence motif found about 15 times per 100 Kb in the human genome [128]. The unique labelling pattern throughout the genome allows for the unambiguous identification of every imaged molecule’s genomic location, resulting in a local consensus map that can be compared to a reference genome to detect structural variants (SVs). The so-called rare variant pipeline is used for this study; it targets mosaic samples and can discover SVs from single molecules across the genome, beginning at 5 Kb and falling to a fraction of 1% in allele frequency. In addition, information on the depth of the genome’s coverage is utilised in order to recognise copy number variants (CNVs) and whole-chromosome aneuploidies [129]. Several recent studies have shown that OGM performs well in the cytogenomic assessment of various haematological malignancies, with a particular emphasis on myeloid neoplasms (acute myeloid leukaemia and myelodysplastic syndromes) and acute lymphoblastic leukaemia cases. In these studies, OGM was able to efficiently detect the bulk of clinically relevant abnormalities reported by standard approaches, while at the same time revealing new cytogenomic information in some situations [130,131]. A cohort study done by Puiggros and her team on 46 CLL patients found that the usage of OGM in CLL enabled them to achieve better characterisation of these patients’ genomic complexity in comparison to current approaches, and also showed increasing detection of cytogenomic abnormalities via the OGM approach which can contribute to adverse disease progression in those CLL patients [103].

NGS genomic oncology profiling assays and GWAS brought into play an unpreceded analytical depth to accommodate the characterisation of the highly complicated genetic landscape of haematological cancers, especially CLL [132], and can become a key driver of personalised cancer care [133]. NGS is able to detect single-nucleotide variants (SNV), small structural changes, and balanced translocations, as well as to confirm CNV detected by array, by providing a base-to-base view of the genome [134] while GWAS is able to identify multiple low-risk variants that together explain about 16% of the familial risk of CLL other than detection of higher-risk SNPs or CNVs associated with disease risk in those families [135]. NGS is also to detect gene mutation in TP53, ATM, NOTCH1, SF3B1, MYD88, and BIRC3; all the aforementioned genes are related to increased susceptibility of patients to develop CLL [58,136,137]. The commonly used NGS platforms are Illumina HiSeq and Illumina MiSeq as well as Ion Torrent from Life Technologies. Data provided by array CGH and NGS technologies has significantly enhanced the knowledge of cancer biology and its underlying driver genes for pharmacogenetics and has guided targeted therapy development and drug-resistance prediction [61].

However, NGS and GWAS has its own pitfalls that need to be addressed. First, the massive amount of data that is obtained from the NGS and GWAS may not be relevant for a diagnostic setting. Second, high cost can be incurred from procurement of NGS equipment, software, and consumables. Third, NGS needs a specialised high-power computer and technician to analyse and store all the data obtained [62]. Increased sensitivity is one of the main benefits of NGS methods for genetic diagnostics; however, so far, this method has only been applied to the detection of single-nucleotide variants (SNVs). Although some chromosomal fusions can be detected using NGS-based approaches with prior knowledge of translocation/fusion partners, a large portion of the genome is still unavailable for structural variant detection due to technical restrictions [138].

A major advantage of using whole-genome sequencing (WGS) is it can identify chromosome inversions and translocation. A study conducted by Robbe et al. (2022) using WGS identified 1248 inversions with frequent breakpoints involving either immunoglobulin light chain kappa (IGK), immunoglobulin heavy chain (IGH) locus, or ch13q14.2 and 993 translocations with no previously documented role in CLL, including t(14;22) with a breakpoint within WDHD1 and t(5;6) (CTNND2-ARHGAP18). Moreover, authors also identified STED2/del3p.21.31, del9p21.3, and gain of chr17q21.31 are associated with relapsed/refractory (R/R) disease and TP53 disruption, whereas MED12 and DDX3X mutations are associated with unmutated IGH CLL [73]. This technology has been reported successfully as not only capturing SNVs with a high level of accuracy but also working well for the detection of disease-causing CNVs. In addition, WGS has the capability of identifying chromosomal rearrangements, as well as STRs and ROH. It is interesting to note that the diagnosis rate of WGS in this study was 27%, which was much higher than the diagnostic rate of clinical microarray (12%) [139].

NGS and arrays are appropriate for cytogenomic studies across a variety of constitutional and cancer research applications, as NGS provides complementary detection capabilities. On a single piece of equipment referred to as the NextSeq 550 System, the researchers are able to carry out both NGS and array scanning. Genome visualisation is possible with the conventional molecular cytogenomic methods for evaluating chromosomal aberrations, such as FISH and karyotyping. However, these approaches often produce a low-resolution image of the genome. As a consequence, the results of such procedures are not always comprehensive [140]. Cytogenomic microarrays provide not only a simple tool but also a reliable method for analysing chromosomal abnormalities at a higher resolution. High-quality microarrays from Illumina are available for the purpose of detecting chromosomal abnormalities while also providing precise and dependable cytogenomic data [140].

In Malaysia, there are a few centres that offers genetic testing in cancer, especially for haematological malignancies. Commonly, most centres will offer CCA and FISH for known chromosomal abnormalities in certain types of haematological malignancies as a tool for diagnosis. They also offer molecular testing (PCR) to detect common fusion genes that are involved in haematological malignancies, such as the BCR-ABL fusion gene in chronic myeloid leukaemia (CML), BCR-ABL fusion gene, TEL-AML1 fusion gene, and E2A-PBX1 fusion gene and MLL gene rearrangement in acute lymphocytic leukaemia (ALL), and PML-RARA gene in acute promyelocytic leukaemia. For array-based technologies such as DNA microarray and NGS, there are not many centres to choose from except for private companies. Furthermore, the array-based technologies are too costly (around ~MYR 2000-MYR 2500 per test) and the need for well-trained staff and experts to interpret the results make them not suitable to be the first-line diagnostic tool in haematological malignancies. However, as the Western countries and other Asian countries such as Korea, China, and Taiwan already used array-based technologies as first diagnostic tools, we need to improve our diagnostic tools so that we are in line with the current diagnosis developments, thus later contributing to better and more precise treatments [79].

This study is a pioneer in Malaysia for performing CLL profiling using a microarray platform using Affymetrix CytoScan 750K array chip; it hopefully will illustrate the genetic aberrations that are involved in CLL pathogenesis. The findings in this study are crucial, as many studies done previously by other populations have already acknowledged the difference of genomic aberrations between Asian CLL and Western CLL [80,141]. Therefore, databases for CLL patients in Malaysia can be created based on these data.

The current assay, called, the “CytoTerra^TM^ Platform”, elevates cytogenetics to new heights. This assay combines the genome-wide structural variation detection capability of conventional cytogenetics with the molecular-level precision of chromosomal microarrays (CMA) and FISH in a single, cost-effective manner with an NGS-based assay. The CytoTerra Platform uses ultra-long-range genome sequencing to assess the breadth of chromosome aberrations with greater resolution than conventional cytogenetic analysis, CMA, and FISH combined. The CytoTerra^TM^ Platform possesses unique features such as genome-wide detection, the ability to detect complex rearrangements, the ability to identify unbalanced chromosomal alterations (deletion, duplication, and amplification), and the ability to examine balanced rearrangements (inversion, insertion, reciprocal, and Robertsonian translocation), and does not require specialised instrumentation [142,143]. Table 4 highlights the advantages and disadvantages of each cytogenomics approach used to diagnose CLL.

All genetic aberration data obtained from CCA, FISH, DNA microarray and whole-genome sequencing in CLL patients will help the clinician to tailor treatment according to patients’ needs, reduce the complication of treatment, and improve survival outcomes [144,145]. Moreover, according to [113], the most recent applications of cytogenomic techniques include conducting research on topologically associated domains, studying interchromosomal interactions, and chromoanagenesis, characterising the 3D structure of chromosomes in various tissue types and shedding light on the multilayer arrangement of chromosomes and the function of repetitive repeats and noncoding RNAs in the human genome.

**Table 4 diagnostics-13-00964-t004:** Cytogenomics approaches for CLL diagnosis.

Technique	Description	Application	Advantage	Disadvantage	Ref.
Conventional cytogenetic analysis (CCA) [G-banding]	Cell culture. The metaphase was treated with trypsin and then stain with Leishman to demonstrate the banding of each chromosome.	Detection of numerical and structural chromosomal abnormalities	Genome-wide screening for chromosomal level anomalies, low cost for reagents and instruments, simple and robust procedures	Low-resolution, required mitotic cells and well spread metaphases, labour-intensive analysis, time-consuming, non-dividing cancer cells cannot be evaluated, poor morphology, insufficient cell for analysis	[104,105]
FISH	Specific probe (DNA fragment) to bind to specific target sequence in chromosome	Identification of the presence, numbers of copies per cell, and localisation of probe DNA, able to detect low level of mosaicism and mosaics of mono- and biallelic deletions	Applicable to interphase cells, fast analysis and scoring, simple and robust procedures	Detection limited to tested target, need specific and reliable reagents, genomic instability (chromothrypsis) and homozygosity (CN-LOH) regions are undetectable, restricted to particular identified genetic regions, relatively expensive and time-consuming due to the fact that each genetic aberration requires its own specific probe, unable to identify any chromosomal abnormalities outside the probe-specified regions of the genome	[47,104,114,115,116,117]
MLPA	Study of several region in the human genome with a single reaction using specific sequence probe	Able to detect genetic aberrations in non-dividing cells with high specificity and sensitivity	High throughput, capable of simultaneously detecting copy number alterations, methylation pattern changes, and/or point mutations in numerous target areas	Cannot detect copy neutral loss of heterozygosity, unable to obtain tumour heterogeneity in low tumour mosaicism, can cross contamination with normal cells, unable to detect balanced translocation, restricted to particular identified genetic regions	[114,115,116,119,120,121]
Array CGH/SNP array (Microarray)	Identification of DNA sequences by specific DNA binding proteins in cells	Identification of cryptic rearrangements (aneuploidy, deletions, duplications, or amplifications), ability to detect copy-neutral loss of heterozygosity (CN-LOH) and some polyploidies	Whole-genome scan, high-resolution target-specific detection (up to > 40kb) of gene amplification, sub microscopic information on imbalances, ability to detect (submicroscopic) areas with genomic instability or chromothripsis, permit a comprehensive screening for copy-number variations (CNAs) over the entire genome in a single experiment	Inability to detect low-level mosaics, insensitivity to heterochromatin, unable to detect balanced translocation, need for well-trained laboratory technologist, high operation costs, poor performance at low tumour levels, failure to detect balanced rearrangements	[50,104,116,127]
NGS (WES&WGS)	Whole-genome analysis	Able to detect single-nucleotide variants (SNV), small structural changes, and balanced translocations as well as to confirm CNV detected by array by providing a base-to-base view of the genome, detection of gene mutation.	High-resolution (covering all coding variation), single-strand sequencing, capable of detecting translocations and inversions of chromosomes	Detection of copy number variant of unknown significance, expensive, need specialised high-power computer and technician to analyse and store all the data obtained	[58,62,72,134,136,137]

## 4. Conclusions

The landscape of CLL genomics will become more thorough and precise with the help of technological evolution. Together with data collected from the DNA microarray technology as well as conventional cytogenetic, FISH, and other advanced technology, whole-genome sequencing may create a new pathway for creating potential therapeutic agents that are more focused on targeted therapy. Despite the fact that there were many methods to detect genomic aberration in CLL, microarray-based technology was deemed to be superior to others (CCA, FISH, MLPA, and PCR) and cost-effective compared to NGS and GWAS. Thus, laboratory technologists should be well-trained and well-versed with microarray technology to keep up with the latest technology. It also helps the clinicians to obtain more detailed data on the disease as well as to determine and quantify disease-associated genetic profiles and improve clinical diagnosis/prognosis, tumour classification, and ultimately, cancer therapy.

## Figures and Tables

**Figure 1 diagnostics-13-00964-f001:**
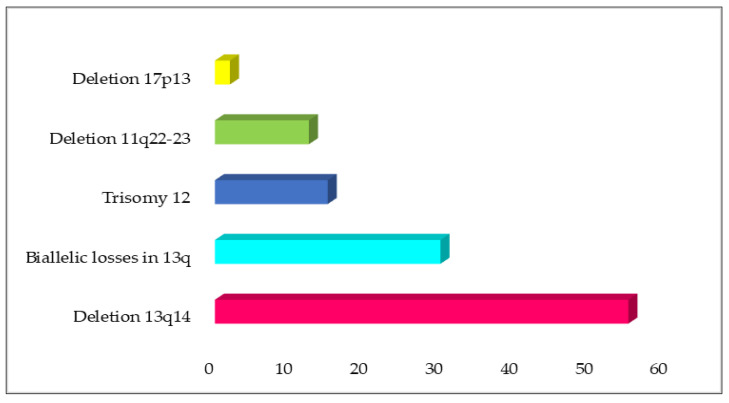
The common chromosomal abnormalities detected in CLL patients by FISH.

**Table 2 diagnostics-13-00964-t002:** The chromosomal abnormalities in CLL patients discovered by various platforms.

Chromosome Aberrations	Prevalence at Diagnosis	Platform	Ref.
Chromosomal translocation	32–42%	Conventional G-banding	[64,65,66]
Complex karyotypes	16%	Conventional G-banding	[64,67]
Deletion in 6q	3–6%	Genomic arrays	[68,69]
Abnormalities in chromosome 8 (8p losses and 8q gains)	2–5%	Genomic arrays	[54]
Deletion in 22q11	15%	Genomic arrays	[70]
Gains of 20q13.12	19%	Genomic arrays	[71]

**Table 3 diagnostics-13-00964-t003:** Predictive biomarkers influence prognosis in CLL.

Gene Mutation	Gene Location	Implication	Prognostic
TP53 mutation	17p13.1	TP53 inactivation causing genomic instability and linked to resistance to radiotherapy and/or chemotherapy.	Poor
NOTCH1 mutation	9q34.3	Act as proto-oncogene which increased the risk for patients to develop Richter syndrome.	Poor
ATM mutation	11q22.3	Dysregulation of cell cycle by impaired detection of DNA damage.	Poor
BIRC3 mutation	11q22.2	Mutation of BIRC3 leads to ligand-independent activation of the constitutive NFκB pathway, inducing cell proliferation and survival.	Poor
IgHV mutation	14q32.33	Mutated IGHV has weaker BCR signalling and results in a higher mutation burden and a lower frequency of driver mutations. It leads to CLL cells proliferate more slowly and less clinically aggressive.	Good
SF3B1 mutation	2q33.1	Mutation of SF3B1 lead to defective RNA messenger splicing and dysregulated cell cycle which leads to rapid disease progression	Poor

## Data Availability

Not applicable.

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
