# Peer review of "Perspectives on the Application of Cytogenomic Approaches in Chronic Lymphocytic Leukaemia"

_diagnostics, 2023, doi:10.3390/diagnostics13050964_

Round 1

Reviewer 1 Report

This review discusses the genetic abnormalities specific to CLL, as well as the use of microarray technology as a diagnostic platform. The topics raised in the work, in my opinion, are very important.
Chronic lymphocytic leukaemia (CLL) is a haematological malignancy characterized by the accumulation of monoclonal mature B lymphocytes (positive for CD5+ and CD23+) in peripheral blood, bone marrow, and lymph nodes. Although CLL is reported to be rare in Asian countries compared to Western countries, the disease course is more aggressive in Asian countries than in its Western counterpart.
The work is supported by extensive literature: 121 items, of which about 43% from the years 2022-2017 and only about 27% from before 2006.

Author Response

Thank you very much for your comments.

Reviewer 2 Report

 Wan Norizzati Wan Mohamad Zamri and colleagues present a review on on chronic lymphocytic leukaemia (CLL) cytogenetic and genetic.

The article is well written and deserves to be published. The topic is also actual.

I would simply suggest to create a couple of table:

1- summarising the different advantages /disadvantages of each technique

2-m  summarising the key cytogenetic feature

These may facilitate the reader 

Author Response

Thank you very much for your comments. Please refer to the attachment.

Reviewer 3 Report

The paper is about Chronic lymphocytic leukaemia (CLL). The paper is most justified given the importance of the problem and the advances in our knowledge and advances in diagnostics. In the introduction, the authors compare the prevalence of the disease in countries as they write of the West to Asian, African and South American countries. I ask at this point for clarification as to which group of countries the authors include Japan. In addition, the authors have forgotten Australia and New Zealand - please supplement the introduction with data from these countries. In the next chapter, the authors characterise the genetics of CCL. Although the authors refer to African and Central American countries in the introduction, they do not mention these populations in this chapter. Since the authors compare Asian and Western populations [156-166] they should also mention Australia, New Zealand, Africa and Central America. Kindly provide information on this matter.

In the next chapter, the authors aptly characterise the huge advances in diagnostics.

The conclusions should be considered appropriate.
The paper is interesting but the authors are very selective in their treatment of individual populations focusing primarily on Asian, European and US residents. The paper should be supplemented with data from other populations.
The literature is good, except for the lack of papers on Australia, for example.

Author Response

(The authors gave the same response as above.)

Round 2

Reviewer 3 Report

once corrected, the paper can be published